# Technical note: Investigation into the relationship between zircon structural damage and Pb mobility using chemical abrasion, SIMS, Raman spectroscopy and atom probe tomography

Charles W. Magee, Jr[1], Lutz Nasdala[2], Renelle Dubosq[3,4], Baptiste Gault[3,5], Simon Bodorkos[1]

[1]Geoscience Australia, Canberra ACT, 2609, Australia
[2]Institut für Mineralogie und Kristallographie, Universität Wien, Josef-Holaubek-Platz 2, A-1090 Vienna, Austria
[3]Max-Planck-Institut für Nachhaltige Materialien GmbH, Düsseldorf, Germany
[4]Department of Earth, Environmental, and Geographic Sciences, The University of British Colombia Okanagan, Kelowna, BC, Canada
[5]Department of Materials, Royal School of Mines, Imperial College, London, UK

*Correspondence to*: Charles Magee (Charles.magee@ga.gov.au)

**Abstract.**

Chemical abrasion (CA), a two-step process of annealing and partial dissolution, is routinely applied to zircon grains prior to U-Pb geochronology, to dissolve portions of the grains affected by Pb loss prior to analysis. Despite the utility of the technique, it is not clear what the more HF-soluble material produced in the annealing step is, what degree of lattice damage causes it to form instead of zircon, how to predict if a specific sub-volume of a zircon will survive CA, or how any of these processes relate to Pb mobility. In this study, we use secondary ion mass spectrometry (SIMS), Raman spectroscopy, and atom probe tomography (APT) to constrain what happens to both concordant and discordant zircon during each step of the CA process. We find that zircon in SIMS sputter craters which have undergone Pb loss generally have more heterogeneous Raman band widths than in those sputter craters where Pb has been retained. Annealing drastically reduces Raman band widths, but some heterogeneity is still present in discordant sputter craters. APT results from all samples which successfully ran were homogenous in U, Pb, Th, and most other elements in all cases. This makes it hard to link Pb loss and lattice damage at the submicrometre scale by direct imaging at this time. However, as the zircon sputter craters with Pb loss show homogenous APT results, we recommend against using homogenous APT results as an indicator of closed-system U-Pb behaviour.

## 1 Introduction

Chemical abrasion is a method used to improve the accuracy and precision of U - Pb dating in zircon by the preferential dissolution of parts of the zircon which have undergone Pb loss. The assumption is that domains of the zircon which have suffered Pb mobility will not anneal back into zircon, but will instead become some other phase which is preferentially dissolved by HF at lower temperatures. The exact nature of the soluble phase formed by annealing is not known, although McKanna et al. (2023) show that it is heterogeneous down to the sub-micrometre limit of their analyses, and Kooymans et al. (2024) show that chemically abraded OG1 zircon has a lower hydroxyl content than untreated OG1 zircon.

McLaren et al. (1994) show that annealing fully metamict zircon at a time and temperature similar to the CA annealing step causes metamict zircon to form discrete $ZrO_2$ and $SiO_2$ phases, which are intergrown on the scale of tens of nanometers. Thus if the material preferentially dissolved in CA undergoes this same decomposition reaction as metamict zircon, it should be visible using a nanoscale imaging technique.

An atom probe is a mass spectrometer with sub-nanometre spatial resolution, in principle with a similar sensitivity to all elements in the range of 10s of ppm (Gault et al. 2010, 2021). Atom Probe Tomography (APT) is the tomographic reconstruction of the analyzed volume based on the temporal and spatial arrival of ions at the detector. This combination of high spatial and chemical resolutions makes it attractive to study property-modifying nanoscale microstructural features. A number of nanofeatures related to various types of alteration have previously been documented in zircon (Valley et al. 2014; Piazolo et al. 2016; Peterman et al. 2016, 2021). Conversely, reference zircons generally produce homogenous results (Exertier et al. 2018; Saxey et al. 2018). This has led some researchers to use a homogenous APT result in zircon as an indicator for closed U-Pb system behaviour (Greer et al. 2023).

Although APT cannot directly image voids, Dubosq et al. (2020) use the electric field distortion caused by voids, and the tomographic reconstruction artifacts generated by this distortion, to detect nanometer-scale fluid inclusions. If the CA dissolution voids observed by McKanna et al. (2023) extend to the nano scale, this should also be visible in APT.

A review about the intersection of radiation dosage, aqueous alteration, lattice damage, and Pb mobility is beyond the scope of this paper. In short, it is not known what combination of radiation damage, aqueous alteration, and other factors induce the crystallographic change to zircon which allows for Pb mobility. Although zircon domains which have lost radiogenic Pb also often gain common lead and light rare earth elements, the extremely low abundance of [204]Pb and lanthanum means that they cannot be used as diagnostic at the sub-micrometre scale, as a nanometer-scale volume is expected to contain less than one atom of ppb level components. The goal of this project was to see if ATP could, in addition to imaging predicted features in annealed and chemically abraded zircon, also find a diagnostic difference in between untreated closed and open system zircon.

## 2. Materials and methods

### 2.1 OG1 target zircon

The OG1 zircon (Stern et al. 2009) was selected for this study due to its well-characterized nature and extensive use as a reference zircon (Stern et al. 2009; Magee et al. 2017; Kemp et al. 2017; Petersson et al. 2019). Measurements of untreated zircons by both TIMS and SIMS identify slight Pb loss, which is ameliorated by chemical abrasion (Stern et al. 2009; Kooymans et al. 2024). This study used archived chemically abraded OG1 (in mount GA5015), and annealed but not partially dissolved material (in mount GA5005). Following crystallization at 3465 Ma, the Owens Gully Diorite underwent regional amphibolite grade metamorphism around 3.3 Ga (François et al. 2014), and the U-Th-He age of the zircons is about 750 Ma (Magee et al. 2017), indicating that it has been cool enough to accumulate radiation damage since at least that time.

## 2.2 Specimen treatment and selection

OG1 zircon grains were annealed in a quartz boat for 48 hours at 1000°C (Bodorkos et al. 2009). The chemically abraded grains were subsequently partially dissolved in HF at 200°C for 10 hours at the Royal Ontario Museum, and the annealed grains had no further treatment. Aliquots of several hundred grains were then mounted in two 25 mm epoxy disks and polished. Both mounts contain the TEMORA-2 reference zircon (Black et al. 2004). Transmitted light, reflected light, and cathodoluminescence images of all zircons were taken prior to SIMS analysis. After the experiments described by Bodorkos et al. (2009), these two mounts were stored in the Geoscience Australia archive before being retrieved for this study.

## 2.3 SIMS U-Pb analysis

Mounts GA5005 and GA5015 were run side-by-side in a single combined SIMS U-Pb session on the Geoscience Australia SHRIMP (Sensitive High Resolution Ion Microprobe) IIe instrument. A primary beam of $O_2^-$ with approximately 10.68 kV impact energy and a primary beam monitor (PBM; net sample current) current of approximately -2 nA was projected through a 100 µm aperture to sputter a flat-bottomed ellipsoidal crater approximately $16 \times 21$ µm across and one µm deep. Analytical techniques were otherwise as per Magee et al. (2012).

## 2.4 Raman spectroscopy

The degree of accumulated radiation damage in zircon was estimated from the full width at half band maximum (FWHM) of the ca. 1000 $cm^{-1}$ Raman band (Nasdala et al. 1995), which is assigned to antisymmetric stretching vibrations of $SiO_4$ tetrahedrons (Dawson et al. 1971). Analyses were done by means of a dispersive Horiba LabRAM HR Evolution spectrometer equipped with Olympus BX-series optical microscope, a diffraction grating with 1800 grooves per mm, and Peltier-cooled charge-coupled device (CCD) detector. Spectra were excited with the 632.8 nm emission of a He-Ne laser (10 mW at the sample surface), using an Olympus 100× objective (numerical aperture 0.90). Wavenumber calibration was done using the Rayleigh line. The system was operated in full confocal mode, resulting in a lateral resolution of better than 1 µm (compare Kim et al. 2020) and a depth resolution of ca. 2 µm. Hyperspectral maps were obtained using a software-controlled x-y stage in "oversampling" mode (step size in the range 0.6-0.9 µm). Fitted FWHM values were corrected for the artefact of experimental band broadening according to the procedure of Váczi (2014), based on the spectrometer's FWHM of the instrumental profile function (IPF) of 0.8 $cm^{-1}$ in the red range. For further details see Zeug et al. (2018).

## 2.5 Atom probe tomography

A suite of six zircon samples were selected for APT analysis, i.e., 05U-16.1 (untreated concordant), 05U-21.1 (untreated discordant), 15U-14.1 (untreated discordant), A-23.2 (annealed concordant), A-17.1 (annealed discordant), and C-13.1 (chemically abraded concordant). A series of specimens (n=4-5) was prepared from each sample by in situ liftout (Thompson et al. 2007). The surface was protected by ion-beam deposition of a ~0.5 µm Pt layer and specimens were shaped into needles

with annular milling at 30 kV on a dual-beam scanning electron microscope/focused ion beam (FEI Helios Nanolab 600i or Helios Plasma-FIB) and subsequently cleaned using the ion-beam at 5 kV to remove regions potentially severely damaged by the implantation of energetic Ga or Xe ions. The specimens were then analyzed by APT in a CAMECA local electrode atom probe (LEAP) 5000 XR fitted with a reflectron lens with a detection efficiency of ~52% at the Max-Planck-Institut für Nachhaltige Materialien, Dusseldorf, Germany (Table S1). The specimens were analyzed at a base temperature of 50-60 K, with a laser pulse energy of 400-700 pJ focused on an area estimated to be <3 μm in diameter, a detection rate of 0.03-0.1 ion detected for 100 pulses and a laser pulse repetition rate of 100-200 kHz. The data processing and reconstruction were done with the commercial software packages AP Suite 6.1 and 6.3. The ranged mass spectrum used for the specimens is shown in Figure S1. A schematic showing the relative analytical volumes of SHRIMP, Raman, and APT is shown in Figure 1.

## 3 Results

### 3.1 SHRIMP U-Pb results

SHRIMP results are shown in Table 1 and Figure 2. They generally agree with similar recent studies from this lab (Magee et al. 2023; Kooymans et al. 2024). Eleven individual SIMS sputter craters (two untreated concordant, two untreated discordant, two annealed concordant, three annealed discordant, and two CA concordant) were chosen for further study using hyperspectral Raman imaging. The U-Pb systematics of these spots are shown in Table 2.

### 3.2 Raman spectroscopy and mapping

The FWHM of the main zircon Raman band near 1000 cm$^{-1}$ is 1.7-1.8 cm$^{-1}$ for well-ordered zircon (Nasdala et al. 2002; Zeug et al. 2018) and increases to well above 30 cm$^{-1}$ at elevated stages of damage accumulation (Nasdala et al. 1995; Zhang et al. 2000). In the present study, FWHM values between 3.2 and 32 cm$^{-1}$ were obtained (Table 3), characterizing the samples as spanning almost the entire range from mildly to severely radiation-damaged. Untreated grains of OG1 are in general moderately to severely radiation-damaged (FWHMs in the range 10-32 cm$^{-1}$), with significant zoning and/or patchy heterogeneity on a scale of within single SHRIMP spots. This is new information, as the OG1 reference zircon has, to the best of our knowledge, never been subjected to any systematic quantitative study of the structural state and its internal heterogeneity, even though cathodoluminescence (CL) images obtained from polished grains indicated some degree of internal structural zonation (Stern et al. 2009; for the dependence of luminescence intensity on radiation damage see Nasdala et al. 2002; Lenz and Nasdala 2015).

Band widths for the annealed and chemically abraded grains (FWHMs 3-12 cm$^{-1}$) were in general much narrower than that of the untreated grains. Based on results of recent annealing studies (Ginster et al. 2019; Ende et al. 2021), the above FWHM values (obtained after samples had experienced 48 h at 1000°C) indicate initial FWHMs of 6-30 cm$^{-1}$ before annealing. There are no systematic differences between structural states of annealed-only and annealed plus chemically abraded samples. Neither are there systematic differences in degrees of radiation damage between concordant and discordant spots, even though

the latter seem to be slightly more heterogeneous. Most of the discordant grains yielded heterogeneous FWHM distribution patterns across the SHRIMP analysis pits (Figure 3). There are generally zones or linear features in the sputter crater maps, showing areas of narrower and wider bands.

### 3.3 APT results

A total of 22 specimens from the six zircon samples were successfully analyzed with APT. All datasets yield similar compositions for zircon major components (i.e., 17.4-21.1 at% Zr; 14.6-17.1 at% Si; and 57.3-62.3 at% O; Table 4). Other element species detected in APT include H, Hf, Y, Pb, Li, Yb, U, Th, Lu, and Tm (Table 4). To evaluate the measurement reliability, the composition of zircon components (Zr, Si, O, H, Y, Pb, Li, Hf, and U) is plotted against the estimated electric field for each dataset (Figure S2). The electric field strength during experiments is approximated using the charge-state ratio of $Si^{2+}/Si^+$ as a proxy. The plots indicating the composition versus field estimate for each of the mentioned species show no apparent trends across datasets from the same sample, thus validating the comparison of the composition obtained from the different experiments. Furthermore, there are no significant trends in compositions between treatment types or Pb retention, indicating similar compositions across samples. The only discernible trend is observed for Y, Pb, and Li, where their compositions are slightly higher in annealed samples compared to chemically abraded and untreated samples. Similarly, the compositions of the same species across different regions of interest within a single specimen (sample 05A-23.2, dataset 78075) was plotted against the estimated field for each region to assess the stability of APT experiments during extended runs (>6hrs; Figure S3). Throughout a single analysis, the electric field displays minimal variation, with $Si^{2+}/Si^+$ ratios ranging from 17.35 to 17.81, while compositions remain consistent despite the minimally fluctuating field.

All datasets had minor 103 Da and 103.5 Da peaks ($^{206}Pb$ and $^{207}Pb$, respectively) above background, however, the $^{208}Pb^{++}$ peak at 104 Da overlaps with $^{28}Si_2{}^{16}O_3{}^+$ (Figure 4, S1, S4). Therefore, only the peaks at 103 and 103.5 Da are considered in the bulk composition analysis and used to evaluate the distribution of Pb in the datasets. The 3D reconstructions of all datasets reveal a homogeneous distribution of Pb and every other major or trace element (Figure S5) with the exception of Li in sample 05A-23.2 (annealed concordant; Figure 5a). A nearest neighbor analysis, which measures the distance between each Pb ion and its closest Pb ion neighbor (d-pair) and plots it against a randomized distribution of ions, confirms the homogeneity of Pb distribution in each sample (Figure 6a-f). The homogeneous distribution of Pb is also confirmed in each sample by performing a radial distribution analysis whereby the bulk normalized composition of each species is plotted as a function of its radial distance to the nearest Pb ion (Figure 6g-l). Since the bulk composition of Pb hovers about a value of 1 for each dataset, homogeneity can be assumed. In sample 05A-23.2 (annealed concordant), the frequency distribution of the d-pair distances for Li is skewed towards lower distances when compared to the randomized distribution curve, suggesting inhomogeneity (Figure 5b). The radial distribution analysis of Li for sample 05A-23.2 reveals elevated bulk normalized compositions of Li for short radial distances, also confirming the heterogeneous distribution of Li in the sample (Figure 5b, 5c).

## 4. Discussion

### 4.1 Raman

Using the measurement parameters detailed in the methods section yielded excellent results despite topographic features of the sputter crater and thin (estimated 20 nm) crater-bottom zone of primary ion-induced lattice damage from the SIMS primary ion bombardment. Untreated discordant zircon tended to have regions where the FWHM exceeded 30 cm$^{-1}$, while in concordant zircon the maximum FWHM was generally less than 25 cm$^{-1}$ (Table 3). This reconfirms that radiation damage lowers the Pb retention performance of zircon. As expected, after annealing (without and with additional CA), zircon recovers much of the radiation-damage, indicated by significantly narrower Raman FWHMs. Here, degrees of damage of discordant and concordant spots overlap, making clear conclusions impossible. Concordant and discordant spots (in both untreated and annealed/CA grains) differ insofar as discordant samples show somewhat higher extents of structural heterogeneity across the SHRIMP analysis pit (Figure 3). Overall, Raman spectroscopy alone does not give conclusive hints to why annealed zircon yields concordant or discordant Pb/U ratios. Raman response of the chemically abraded samples were similar to the annealed ones in terms of band width, and had few features. This technique could potentially be used prior to spatially-resolved U-Pb analyses, although the time and potential expense involved would probably limit such use to very valuable samples.

### 4.2 Nanoscale element distribution

Elements detected include O, Si, Zr, H, Hf, Y, Pb, Li, Yb, Lu, Tm, U, and Th. Hydrogen is considered to be a contaminant from the vacuum. Nearly all zircon components revealed a homogenous distribution in all specimens, with the single exception of Li in 05A-23.2. Aluminium was not detected. This is not surprising given the near-mantle O isotope values of OG1 (Avila et al. 2020), which indicate minimal crustal assimilation by the parental magma, and the low (< 3 ppm) Al concentrations recorded in previous studies of OG1 zircon (Kooymans et al. 2024). In contrast, yttrium is abundant in OG1 zircons, with Kooymans et al. (2024) reporting median values in excess of 1000 ppm. None-the-less, there is no evidence of clumping of Y and / or Pb in any of the analysed APT specimens. This suggests that the regional amphibolite grade metamorphism (François et al. 2014) which gives the Owens Gully Diorite its foliation was not hot or long enough to initiate the Pb ± Y ± Al clump formation seen in zircons subject to hotter granulite or UHT conditions (Peterman et al. 2016; Piazolo et al. 2016). Phosphorus was not detected; spatial correlation between Y and P was not possible.

Since all elemental distributions are homogenous across all specimens, no discernible differences can be attributed to the treatment types at the tens to hundreds of nm scale of APT. Similarly, no differences are observed between discordant and concordant grains. SHRIMP analyses with up to 9% discordance yield APT results that are homogenous and identical to concordant grains, despite being from areas with the highest levels of lattice damage, as shown by Raman spectroscopy. Potential explanations for the homogeneous distribution of elements are discussed below. Based on these data, we suggest that APT not be used as the sole method for determining closure in the U-Pb system for zircon.

McKanna et al. (2023) show that CA produces voids down to the sub-micrometre limit of their spatial resolution. Wang et al. (2020) demonstrate that APT can detect voids in metal alloys in the form of artefacts showing apparent clustering or dispersion of matrix ions due to trajectory aberrations during evaporation caused by the distortion of the electrostatic field created by the void. Dubosq et al. (2020) demonstrate that this technique can also be used in geologic materials by identifying voids in the form of fluid inclusions in Archean pyrite. As all elements in the atom probe results from the chemically abraded sample (C-13.1) are homogenous, we have no evidence that CA produces voids at the nanometre scale of Dubosq et al. (2020)'s fluid inclusions.

The homogenous distribution of Si and Zr in the annealed samples (A-17.1, A-23.2) suggests that the formation of amorphous silica and $ZrO_2$ through annealing metamict zircon at similar temperatures, as demonstrated by McLaren et al. (1994) has not occurred in any of the eight ATP tips successfully run from these samples.

## 4.3 Potential Mechanisms of Pb loss

Micrometre-to submicrometre-scale heterogeneity needs to be considered as one feature potentially favouring secondary loss of radiogenic Pb. Heterogeneous volume expansion of neighbouring sample volumes results in complex strain patterns and, if the elasticity maximum is exceeded, the opening of fractures that then may serve as ideal pathways for fluids (Peterman et al. 1986; Chakoumakos et al. 1987; Lee and Tromp 1995; Nasdala et al. 2010). However, on the resolution scale of a powerful optical microscope, such fractures were not always observed. Nonetheless, McKanna et al. (2023; 2024) show that chemical abrasion selectively dissolves zircon in such a manner down to the limit of their imaging resolution (about half a micrometre) and that the leachate solutions from the CA partial dissolution have disturbed U-Pb systematics.

If the discordant areas in zircon are discrete areas of damage distributed within a matrix of isotopically closed zircon, there is a possibility that sampling might miss these regions purely by chance. The likelihood of missing the damaged areas by chance depends on distribution of Pb loss. If disturbed regions are a mix of intact and damaged areas, the probability of randomly sampling only the intact areas is low, calculated as (1-damaged fraction) ^ n. Considering a SHRIMP spot with 5% Pb loss, where the Raman can exclude 75% of the spot area as having lost Pb, the remaining 25% of the spot would then account for the Pb loss. Given this scenario, the remaining area would have experienced a 20% Pb loss. The most extreme possibility is that 20% of this area has lost 100% of its Pb, and the remaining 80% is concordant. As this study had nine successful APT analyses (on three discordant grains), the probability of choosing concordant areas by chance is $0.8^9$, or 13%. It is important to note that this is a maximum, assuming total Pb loss from the altered areas. McKanna et al. (2024) show that the areas altered enough for chemical abrasion to dissolve them still contain significant Pb. If instead we assume that 40% of the area has suffered 50% Pb loss, then the probability of sampling only the intact areas drops to 1% ($0.6^9$).

Of course, these estimates assume that APT analytical volumes can be chosen and run without bias. It is interesting that of the eight specimens which failed while being run in the Atom Probe, six were from discordant SHRIMP spots. As the accumulation of radiation damage involves substantial volume increase, heterogeneous self-irradiation of zoned zircon will necessarily result in complex strain patterns (compressive in more, and dilative in less damaged volumes). Fracture of APT

specimens is generally assumed to be caused by the electrostatic pressure arising from the intense electrostatic field used to field evaporate the surface atoms over the course of the experiment, making brittle materials more prone to early fracture (Wilkes et al. 1972). If strained polyphase material leads to an enhanced propensity for specimen failure before any APT data can be acquired, then there will be a strong selection bias in the results favouring intact zircon. We are not aware of any successful APT analyses of metamict zircon in the literature, either on purpose or by accident. While the inherited cores analysed by Peterman et al. (2016) show Pb loss, that material was reannealed in Mesozoic granulite facies metamorphism, and has had minimal radiation damage accumulation since then. It is possible that the APT analytical technique gives a Panglossian view of zircon crystal chemistry, by destroying badly disturbed samples instead of analysing them.

## 5. Conclusions

The present study shows that discordant SHRIMP spot analyses were obtained from severely radiation-damaged spots (as indicated by strongly broadened Raman bands). Annealing the zircon samples reduces the band widths in both discordant and concordant zircon, while discordant spots have more residual heterogeneity in their Raman maps. Never-the-less, APT analysis of concordant, discordant, untreated, annealed, and chemically abraded OG1 all show homogenous distribution of all detected major and minor elements, including U and Pb. As a result, APT cannot identify any differences between concordant SHRIMP spots and those which are up to 7% discordant. We therefore recommend that APT not be used as a method for determining closure in the U-Pb system for zircon. The similarity between APT results from untreated, annealed, and chemically abraded zircon is difficult to interpret, but does show that the Pb and Y cluster formation found in high temperature metamorphism and experimental tests does not appear at the temperatures and times used for chemical abrasion.

## Author contributions

CM and SB designed and initiated the experiment. CM did the SIMS analysis. LN did the Raman analysis. RD and BG did the Atom Probe analysis. All authors contributed to the text, figures, and tables.

## Acknowledgements

We thank Geoff Fraser and Antony Burnham, Don Davis, and Luke Daly for their thoughtful reviews. This paper is published with the permission of the CEO, Geoscience Australia.

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

**Figures and captions:**

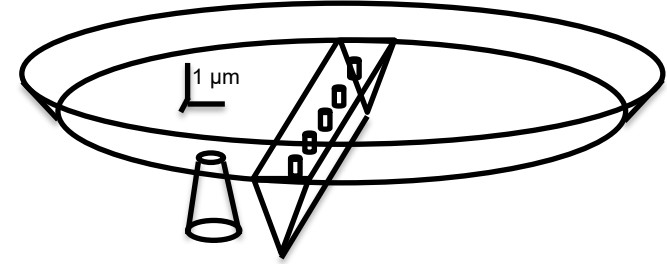

**Figure 1: A scaled drawing showing the relative analytical volume of the SHRIMP, Raman, and APT analytical techniques. Raman and APT volumes are shown in the bottom of a SHRIMP sputter crater, as per this study. SHRIMP sputter crater is approximately 17x22x1 µm. Raman excitation volume is approximately 2 µm deep and 1 µm across. APT ion milled area removed is about 20x2x2 µm, while the individual samples produced are about 0.5 µm x 0.1 µm x 0.1 µm. The SHRIMP and APT are destructive, while the Raman is not.**

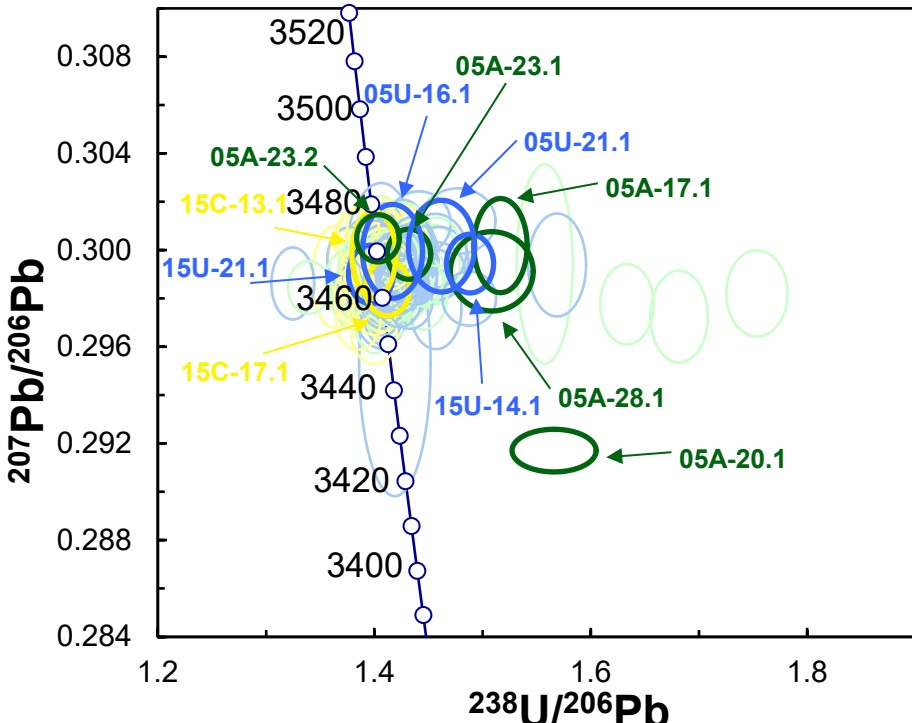

**Figure 2: Concordia diagram of OG1 SHRIMP geochronology results. Untreated zircons are blue. Annealed zircons are green. Chemically abraded zircons are yellow. Analytical spots selected for APT analyses are highlighted and labelled. Data-point error ellipses represent the 68.3% confidence level.**

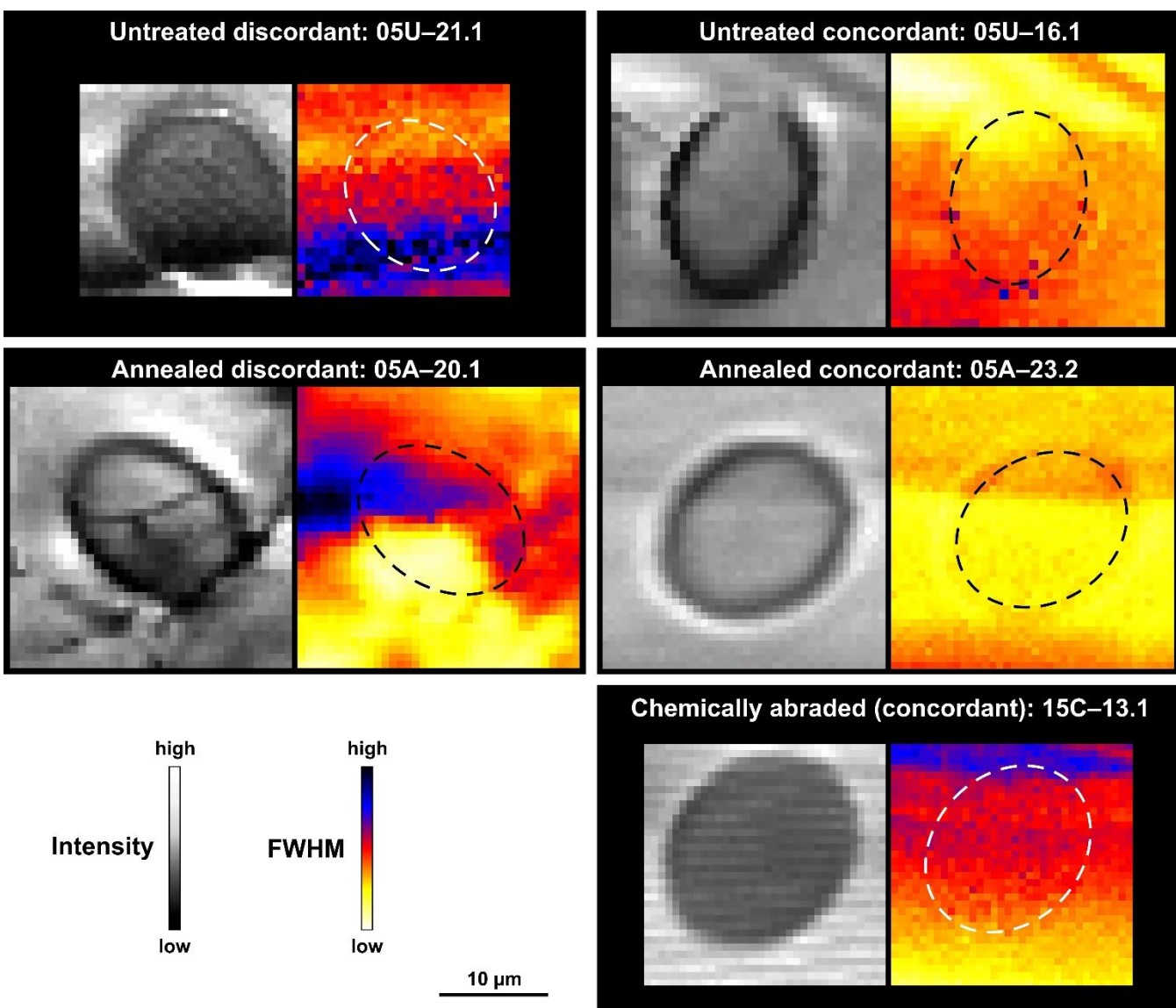

**Figure 3: Pairs of Raman maps (step sizes in the range 0.6-0.9 μm) obtained from selected SHRIMP analysis pits, showing distribution patterns of fitted parameters of the main v₃(SiO₄) zircon band near 1000 cm⁻¹. Left, intensity on an arbitrary grayscale, visualising the locations of spots. Right, FWHM (the locations of SHRIMP pits are visualised by dashed ovals). Colour-coded FWHM ranges (given in cm⁻¹) are 10-28 (05U-21.1; 05U-16.1), 4-12 (05A-20.1), 3-7 (05A-23.2) and 4-9 (15C-13.1), respectively.**

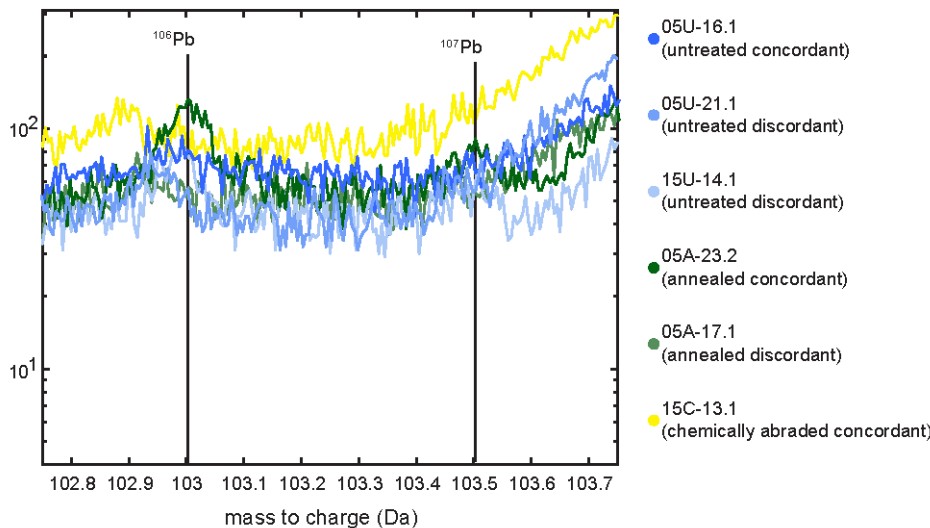

**Figure 4: $^{206}$Pb and $^{207}$Pb spectra from each sample.**

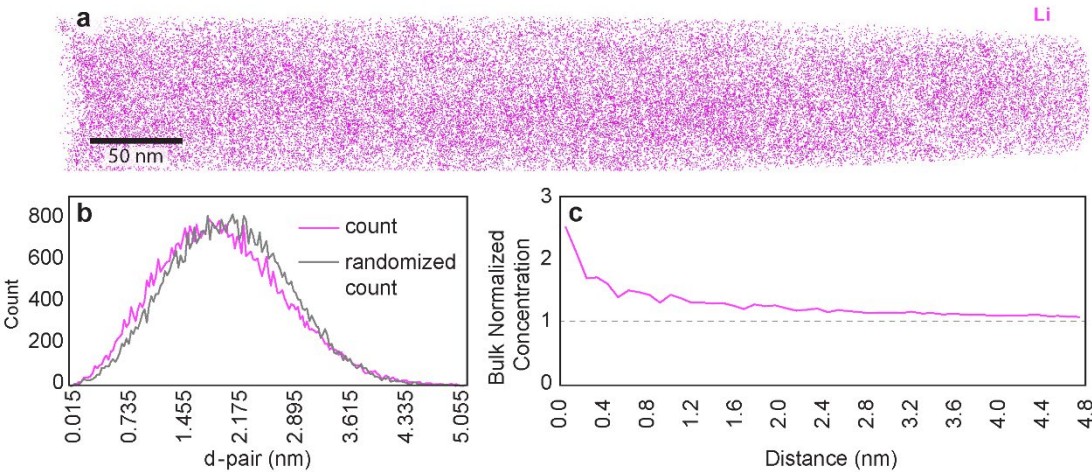

**Figure 5. Li distribution analysis. Sample 05A-23.2. a: Li atom locations in tomographic reconstruction. b: Comparison of observed (pink) and expected (black) d-pair distance. c: Bulk normalized concentration of Li versus distance.**

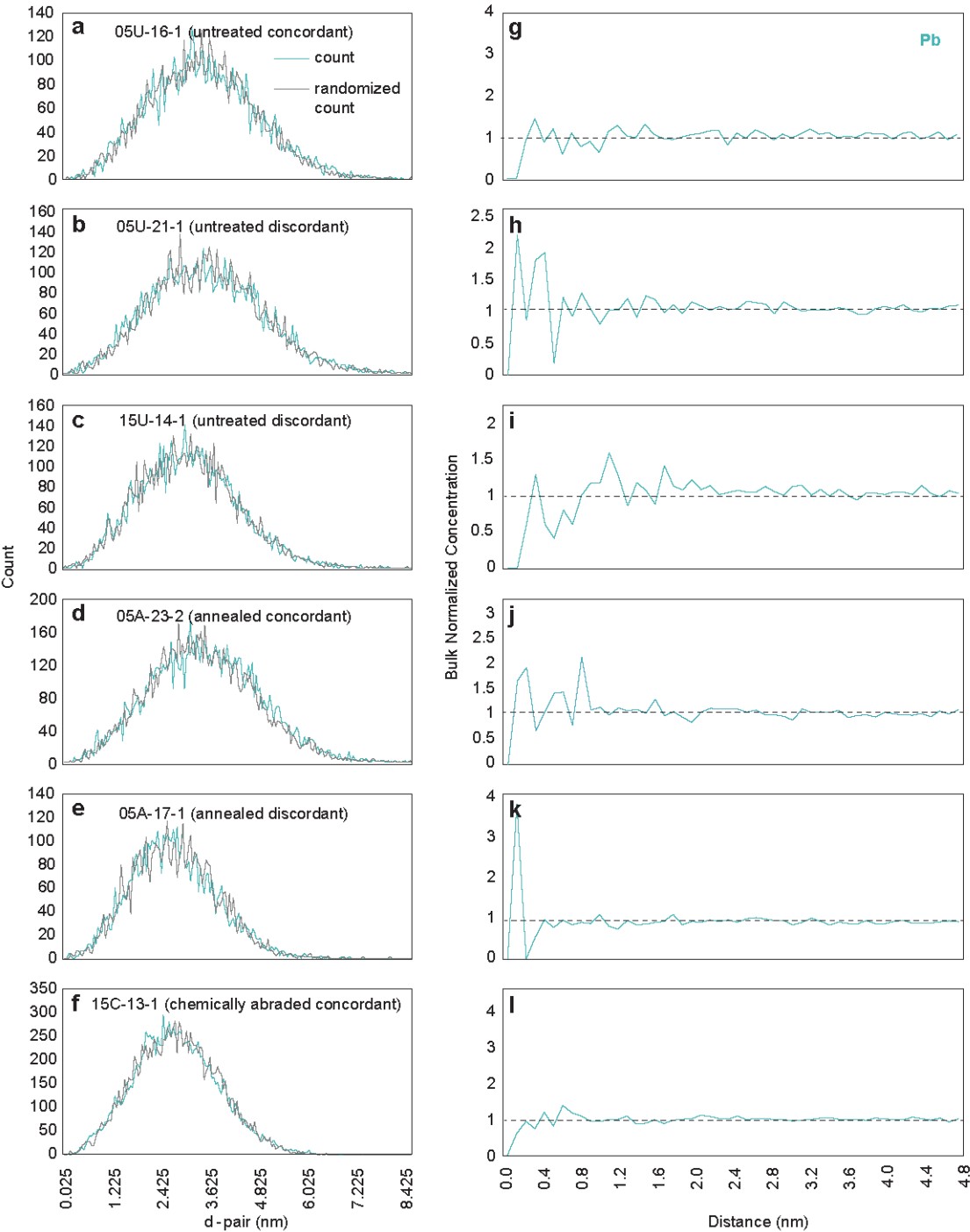

**Figure 6. Pb distribution analysis. a-f count vs randomly generated count vs nearest neighbour distance. g-l: difference between random and measured counts vs distance.**

# Tables

## Table 1. Population weighted mean geochronology results for OG1 analyses.

| Mount | sample | $^{206}Pb/^{238}U$ ratio | Inter 2σ% | Exter 2σ% | $^{206}Pb/^{238}U$ Date | ±X 2σ Ma | ±Y 2σ Ma | MSWD | PoF | n (rej) | $^{207}Pb/^{206}Pb$ ratio | 2σ er % | $^{207}Pb/^{206}Pb$ date | er, 2σ Ma | MSWD | PoF | n (rej) |
|---|---|---|---|---|---|---|---|---|---|---|---|---|---|---|---|---|---|
| GA5005 | Untreated | 0.705094 | 0.37 | 0.46 | 3439.9 | 9.9 | 12.5 | 0.74 | 0.82 | 40 (5) | 0.29923 | 0.09 | 3466.2 | 1.5 | 0.85 | 0.73 | 40 (0) |
| GA5015 | Untreated | 0.704382 | 0.32 | 0.53 | 3437.2 | 8.7 | 11.6 | 0.88 | 0.74 | 41 (3) | 0.29922 | 0.08 | 3466.2 | 1.3 | 0.73 | 0.90 | 41 (0) |
| GA5005 | Annealed | 0.711277 | 0.40 | 0.49 | 3463.3 | 10.8 | 13.2 | 0.55 | 0.94 | 40 (10) | 0.29908 | 0.09 | 3465.4 | 1.5 | 0.98 | 0.50 | 40 (1) |
| GA5015 | CA | 0.712704 | 0.32 | 0.43 | 3468.6 | 8.8 | 11.6 | 0.55 | 0.96 | 40 (0) | 0.29933 | 0.09 | 3466.8 | 1.4 | 1.23 | 0.16 | 40 (1) |

MSWD= mean squared weighted deviation ($\chi^2/\upsilon$); PoF = probability of fit. Uncertainties are x/y as per Schoene et al. (2006)

## Table 2. Geochronology results for spots subject to Raman mapping.

| Mount | Spot | f 206 * | U (ppm) | Th (ppm) | $^{232}Th/^{238}U$ | $^{204}Pb$-corr $^{238}U/^{206}Pb*$ | err (%) | $^{204}Pb$-corr $^{207}Ob*/^{206}Pb*$ | err (%) | $^{204}Pb$-corr $^{206}Pb/^{238}U$ age (Ma) | 1s err (Ma) | $^{204}Pb$-corr $^{207}Pb/^{206}Pb$ age (Ma) | 1s err (Ma) | Disc.* * (%) | Type |
|---|---|---|---|---|---|---|---|---|---|---|---|---|---|---|---|
| GA5015 | 15U-21.1 | 0.01 | 185 | 82 | 0.46 | 1.3968 | 1.0 | 0.29899 | 0.3 | 3481 | 26 | 3465 | 4 | -0.6 | Untreated concordant |
| *GA5005* | *05U-16.1* | *0.12* | *203* | *116* | *0.59* | *1.4169* | *1.3* | *0.29995* | *0.4* | *3443* | *35* | *3470* | *7* | *1.0* | *Untreated concordant* |
| *GA5015* | *15U-14.1* | *-0.01* | *201* | *98* | *0.50* | *1.4885* | *1.0* | *0.29944* | *0.3* | *3313* | *25* | *3467* | *4* | *5.7* | *Untreated discordant* |
| *GA5005* | *05U-21.1* | *0.00* | *380* | *274* | *0.74* | *1.4621* | *1.4* | *0.30017* | *0.4* | *3359* | *36* | *3471* | *6* | *4.1* | *Untreated discordant* |
| GA5005 | A-23.1 | 0.29 | 265 | 340 | 1.32 | 1.4321 | 0.9 | 0.29983 | 0.2 | 3414 | 24 | 3469 | 4 | 2.1 | Annealed concordant |
| *GA5005* | *A-23.2* | *-0.01* | *247* | *299* | *1.25* | *1.4034* | *0.9* | *0.30049* | *0.2* | *3468* | *25* | *3473* | *3* | *0.2* | *Annealed concordant* |
| GA5005 | A-20.1 | 0.10 | 340 | 136 | 0.41 | 1.5661 | 1.6 | 0.29171 | 0.2 | 3183 | 41 | 3427 | 3 | 9.0 | Annealed discordant |
| *GA5005* | *A-17.1* | *0.19* | *221* | *204* | *0.95* | *1.5166* | *1.1* | *0.30019* | *0.4* | *3265* | *27* | *3471* | *7* | *7.6* | *Annealed discordant* |
| GA5005 | A-28.1 | 0.10 | 225 | 142 | 0.65 | 1.5086 | 1.7 | 0.29912 | 0.4 | 3278 | 43 | 3466 | 6 | 6.9 | Annealed discordant |
| *GA5015* | *C-13.1* | *0.03* | *202* | *166* | *0.85* | *1.3996* | *1.0* | *0.29981* | *0.4* | *3475* | *26* | *3469* | *6* | *-0.2* | *CA concordant* |
| GA5015 | C-17.1 | 0.00 | 197 | 158 | 0.83 | 1.4122 | 1.0 | | 0.3 | 3451 | 26 | 3462 | 4 | 0.4 | CA concordant |

Note: Samples in italics were selected for APT analysis after Raman mapping.

* f 206 = common $^{206}Pb$ / total $^{206}Pb$, calculated from the observed $^{204}Pb$.

** U–Pb discordance; difference between the $^{204}Pb$-corrected $^{206}Pb/^{238}U$ and $^{207}Pb/^{206}Pb$ ages.

## Table 3. Raman results obtained in hyperspectral x-y maps.

| Spot | Type | General description | $\nu_3(SiO_4)$ Raman band Raman shift (cm$^{-1}$) | FWHM (cm$^{-1}$) |
|---|---|---|---|---|
| 15U-21.1 | Untreated concordant | Moderate radiation damage, no fine-scale zoning | 999.3-1001.8 | 10.2-19 |
| *05U-16.1* | *Untreated concordant* | *Moderate radiation damage, no fine-scale zoning* | *998.8-1000.1* | *12.5-22* |
| *15U-14.1* | *Untreated discordant* | *Moderate to severe radiation damage, no fine-scale zoning* | *997.0-1000.5* | *15-32* |
| *05U-21.1* | *Untreated discordant* | *Moderate to severe radiation damage, no fine-scale zoning* | *998.2-1000.4* | *15-30* |

| | | | | |
|---|---|---|---|---|
| A-23.1 | Annealed concordant | Mildly to moderately damaged, patchy zoning | 1005.0-1007.2 | 4.5-9.2 |
| *A-23.2* | *Annealed concordant* | *Mildly damaged, weak primary zoning* | *1006.7-1007.3* | *3.7-4.7* |
| A-20.1 | Annealed discordant | Mildly to moderately damaged, fracture, patchy zoning | 1004.0-1007.1 | 4.6-11.9 |
| *A-17.1* | *Annealed discordant* | *Mildly damaged, weak primary zoning* | *1007.1-1007.7* | *3.2-5.4* |
| A-28.1 | Annealed discordant | Mildly damaged, weak primary zoning | 1005.1-1006.1 | 5.5-7.6 |
| *C-13.1* | *CA (concordant)* | *Mildly damaged, weak primary zoning* | *1004.9-1005.9* | *5.5-7.7* |
| C-17.1 | CA (concordant) | Mildly damaged, weak primary zoning | 1005.4-1006.1 | 4.8-7.2 |

Note: Samples in italics were selected for APT analysis.

**Table 4. Summary of APT results**

**Table 4. APT bulk composition analysis (at%)**

| Dataset | 05U-16-1 (untreated concordant) | | | 05U-21-1 (untreated discordant) | | | 15U-14-1 (untreated discordant) | | | A-23-2 (annealed concordant) | | | | | A-17-1 (annealed discordant) | | | C-13-1 (chemically abraded) | | | | |
|---|---|---|---|---|---|---|---|---|---|---|---|---|---|---|---|---|---|---|---|---|---|---|
| | 84840 | 84845 | 85062 | 82932 | 82937 | 83650 | 87032 | 87058 | 87473 | 78075 | 78174 | 78229 | 78402 | 78434 | 79815 | 79906 | 80175 | 85476 | 85586 | 85640 | 85690 | 85698 |
| H | 5.785 | 6.529 | 7.210 | 3.467 | 5.261 | 5.010 | 1.712 | 2.287 | 1.138 | 2.309 | 3.389 | 2.118 | 5.845 | 6.762 | 5.922 | 9.323 | 8.555 | 5.245 | 8.827 | 10.83 | 4.459 | 3.386 |
| O | 58.81 | 57.86 | 58.07 | 60.22 | 59.81 | 60.54 | 61.14 | 62.27 | 61.71 | 60.54 | 59.66 | 60.28 | 57.47 | 57.45 | 58.75 | 57.28 | 57.56 | 58.37 | 56.86 | 55.56 | 59.62 | 60.20 |
| Zr | 19.90 | 19.99 | 19.53 | 20.30 | 19.25 | 19.21 | 20.59 | 19.25 | 20.52 | 20.71 | 20.77 | 21.06 | 20.62 | 19.81 | 19.70 | 18.54 | 18.66 | 20.09 | 18.85 | 17.43 | 18.68 | 19.90 |
| Si | 15.26 | 15.48 | 14.98 | 15.84 | 15.50 | 15.02 | 16.26 | 15.94 | 16.48 | 16.08 | 15.93 | 16.27 | 15.82 | 15.74 | 15.37 | 14.64 | 14.96 | 16.03 | 15.26 | 16.00 | 17.06 | 16.26 |
| Y | 0.046 | 0.039 | 0.043 | 0.032 | 0.026 | 0.028 | 0.033 | 0.026 | 0.020 | 0.111 | 0.080 | 0.087 | 0.077 | 0.081 | 0.071 | 0.068 | 0.070 | 0.034 | 0.025 | 0.025 | 0.036 | 0.052 |
| Pb | 0.017 | 0.001 | 0.002 | 0.003 | 0.007 | 0.000 | 0.008 | 0.004 | 0.005 | 0.006 | 0.005 | 0.003 | 0.003 | 0.009 | 0.009 | 0.007 | 0.001 | 0.004 | 0.006 | 0.003 | 0.005 | 0.007 |
| Li | 0.014 | 0.002 | 0.002 | 0.012 | 0.011 | 0.007 | 0.047 | 0.019 | 0.009 | 0.054 | 0.027 | 0.026 | 0.028 | 0.014 | 0.022 | 0.021 | 0.024 | 0.030 | 0.020 | 0.024 | 0.024 | 0.027 |
| Th | 0.000 | 0.001 | 0.000 | 0.000 | 0.000 | 0.004 | 0.000 | 0.001 | 0.000 | 0.000 | 0.003 | 0.000 | 0.004 | 0.000 | 0.007 | 0.002 | 0.000 | 0.004 | 0.001 | 0.001 | 0.000 | 0.000 |
| Yb | 0.007 | 0.004 | 0.007 | 0.004 | 0.003 | 0.004 | 0.009 | 0.004 | 0.006 | 0.008 | 0.005 | 0.006 | 0.003 | 0.004 | 0.006 | 0.002 | 0.006 | 0.008 | 0.004 | 0.001 | 0.002 | 0.006 |
| Lu | 0.000 | 0.001 | 0.001 | 0.001 | 0.000 | 0.004 | 0.001 | 0.004 | 0.003 | 0.002 | 0.001 | 0.002 | 0.002 | 0.001 | 0.001 | 0.001 | 0.001 | 0.001 | 0.003 | 0.002 | 0.003 | 0.001 |
| Hf | 0.168 | 0.103 | 0.153 | 0.126 | 0.129 | 0.164 | 0.195 | 0.190 | 0.123 | 0.171 | 0.133 | 0.155 | 0.121 | 0.127 | 0.135 | 0.111 | 0.162 | 0.185 | 0.147 | 0.132 | 0.123 | 0.153 |
| U | 0.005 | 0.002 | 0.001 | 0.001 | 0.001 | 0.002 | 0.003 | 0.002 | 0.001 | 0.002 | 0.001 | 0.001 | 0.001 | 0.000 | 0.001 | 0.005 | 0.002 | 0.002 | 0.002 | 0.001 | 0.002 | 0.003 |
| Tm | 0.001 | 0.002 | 0.002 | 0.000 | 0.001 | 0.001 | 0.001 | 0.001 | 0.000 | 0.002 | 0.001 | 0.002 | 0.001 | 0.001 | 0.001 | 0.002 | 0.001 | 0.000 | 0.002 | 0.001 | 0.002 | 0.002 |
| Si2+/Si+ | 19.39 | 24.60 | 22.99 | 25.20 | 25.53 | 27.93 | 20.44 | 22.46 | 23.14 | 20.00 | 19.36 | 21.05 | 18.25 | 17.07 | 23.35 | 20.34 | 24.38 | 22.68 | 23.88 | 21.93 | 24.37 | 25.41 |
