# Peer review of "Technical note: Investigation into the relationship between zircon structural damage and Pb mobility using chemical abrasion, SIMS, Raman spectroscopy and atom probe tomography"

_EGUsphere, 2025_

## Author Response (AR1)

Response to editor comments:

Major edits:

*Abstract and introduction were both rewritten and drastically shortened.*

*Sample description was shortened.*

*SHRIMP methodology was shortened.*

*SHRIMP results were shortened.*

*SHRIMP discussion deleted.*

*Homogenous Pb loss discussion section deleted.*

*Editor's suggestion to not add material asked for by the reviewers has also been taken (but we have reworked figure 1 for clarity).*

Minor edits line-by-line:

Title:
*Rewritten.*

Abstract:*Rewritten.*

Section 2.3+2.4+2.5: I would like you to strongly consider referencing a recent, previously published reference for the analytical techniques here instead of writing them out in full. The analyses are not made in an extraordinary manner, and readers are likely to consider them fit-for-purpose without having a full explanation here.

*Done for section 2.3.*

*We have left section 2.4 because this has never actually been done before, and it's easy to generate artifacts by doing zircon Raman badly.*

*For section 2.5, the analytical details are pertinent to the non-detection of heterogeneity by APT, so we have left this and cut extraneous parts of the discussion instead.*

Section 3.1: Please strongly consider abbreviating this section and concatenating it with section 4.1. The results are not the focus of the study, and they are, as you say, essentially identical to previously published results by your group, and essentially as-expected.

*Done.*

Line 286-287: There is no evidence, here or elsewhere, that volume diffusion of Pb is an important process in Pb loss. Please delete this line.
Line 288: This line is written in such a way that it implies that in the work of McKanna et al., that portions of zircon dissolved in CA = portions of zircon that have had Pb-loss. Please here (and elsewhere that this study is described) clarify for a non-specialist reader that the dissolved portions of zircon in the McKanna et al. work include portions of zircon that were closed U/Pb systems.

*We have removed this entire section, mostly for these reasons and to shorten the paper.*

Reviewer 1 line-by-line:

Line 35: 'vanishingly', non-scientific term.

*Removed in rewrite*

Line 139-140: Regarding the common Pb correction in SHRIMP, there should be no significant primary common Pb in zircon, especially not for near-concordant early Archean aged grains. As I understand it, the small 204Pb signal measured by SHRIMP is derived from Pb in the Au coating, which should have a fixed isotopic composition.

*Our laboratory blank is in the hundreds of zeptomole range (Ickert et al. 2013 Appendix), and Kooymans et al. (2024) show that co-mounted untreated and chemically abraded zircons have different (sometimes much different) common Pb contents.*

Line 238-239: 'This reconfirms that radiation damage lowers the Pb retention performance of zircon.' This statement seems to imply that radiation damage leads to discordance, for which I see no direct evidence. It is a pre-condition necessary for low temperature Pb loss following alteration. It might have been worthwhile to search for any evidence of alteration using high resolution BSE images of the SHRIMP target areas associated with discordant data.

*Yes, but BSE wouldn't have told us what the alteration is; it would only tell us about a change in density or mean atomic number. Our Raman mapping gave us more information. Also, we have a wide variety of additional images which has not been included for space, including the SEM images related to preparing the APT sample. For example, this is what our CA OG1 zircon (spot C-13.1) looks like at 3500x with the protective metal foil added:*

[Figure]

Line 276: 'We know from the U-Pb data that the discordant spots in this study have lost Pb.' This is a tautology.

*Section removed.*

Line 277: '…these data'.

*Section removed.*

Line 286-287: 'Our results are hence consistent with radiation-damage-enhanced volume diffusion of Pb out of the zircon.' Same comment as for Line 238-239 above. If radiation damage did lead to diffusive loss of Pb then it is hard for me to see how annealing damaged zones followed by leaching would consistently remove discordant domains while partially preserving those with damage. Alteration is very soluble in HF and should be selectively removed.

*Section removed. In a broader sense, we agree that volume diffusion of Pb out of intact zircon is unlikely, but we were trying to express that our results don't provide additional support for why the mechanism is almost certainly something else.*

Fig. 1 This figure is confusing and could be composed with more care. It is hard to associate subfigures with their labels. What I presume is the scale bars look like a sub-figure.

*This figure has been cleaned up and simplified.*

Fig. 2 Again, more care should be put into this figure. The colours for same of the groupings make the ellipses virtually invisible. The figure contains a great deal of information that might be easier to absorb using a legend rather than having to read the caption to interpret which points are associated with which process.

*We agree that the colours aren't great, but given the short timeframe of revision, we were not able to agree on a colour scheme that is better for both dichromatic and trichromatic readers.*

Fig 3: Along with the textural maps in Fig 3, it would be useful to show high resolution CL and BSE images of both the spots and their whole grains before SHRIMP analysis to see the context of the spots.

*We have chosen not to add to the manuscript as per the editor's instructions, but in general we targeted areas of greater Raman band width, if they existed in the crater floor, and chose SHRIMP spot locations based on transmitted light, reflected light, and cathodoluminescence images of all mounted zircons.*

Reviewer #2 line by line:

Line 71-73. It was not clear to me how the sentence regarding baddeleyite flows on from the preceding sentence and seems out of place. This paragraph should be refined to make the relevance of the ionization efficiency and orientation dependance in baddeleyite to the present study clear.

*The reverse discordance observed by McLaren et al. (1994) in metamict zircon which had been annealed into amorphous SiO2 + ZrO2 was published before the problems with measuring elemental U-Pb ratios in ZrO2 were documented by Wingate & Compston (2000), So the McLaren et al. (1994) SHRIMP results should be re-interpreted with the understanding that ZrO2 behaves badly in SIMS. But we've deleted this section anyway.*

Line 200. It was unclear to me what constituted a successful atom probe analysis as no information was given as to dataset size. I'm not sure if supplementary materials are permitted but, if possible, it would be worth adding a table in line with Blum et al's 2018 recommendations on the best practices for reporting atom probe analyses of geological materials.

*The supplementary materials are available for download, and this is table S1.*

Line 214-216. Pb isotope peaks were detectable. However, were corresponding U peaks also observable in the APT datasets? It is hinted that this is the case later in the manuscript but if so these should also be presented.

*The U+++ peak at 79.333 is shown in figure S1.*

Line 216. It is very interesting that all trace elements are homogenous except for Li but might be worth explicitly mentioning Y and Al which formed clusters in previous studies. I note that the absence of Al and evaluation of Y is noted later in the discussion but may worth briefly noting in the results as well.

*Yttrium was noted in line 208 (now line 136)*

Line 329. 'APT not be used for determining closure' despite my earlier comment regarding making this point more strongly I think here it needs the caveat of APT not be used in isolation or as the sole method used for determining closure. I think it would also be worth adding a recommendation of combining APT with CL/Raman/EBSD to determine potential alteration/discordance.

*The APT in this study did not provide any additional information, and thus we can't recommend its use in this context.*

Figure 3. There are clear zones within the Raman data within the SHRIMP pits of the discordant zircons, if possible, would the authors be able to show where each APT dataset was taken from within the pit and which ones failed? Were they at the interface or within one or other of the domains.

*We targeted maximum Raman band width areas where present, subject to technical constraints,*

Figure 4. Add the U peaks to the figure.

*The U peaks are in the supplement, We concentrate on Pb in the main paper, as Pb is the element that mobilizes in the vast majority of open system zircons.*

Table 4. If possible, add U/Pb isotopes ratios to the table as well as model ages.

*The precision of the calculations we did was far too low to be useful, and the best method of defining and measuring U-Pb ratios in APT data is beyond the scope of this paper. For example, the Pb/U ratios in the samples from 05U-21.1 range from 0 to 7, which correspond to ages from the present day to before the formation of the Earth.*